materials science/environmental chemistry/inorganic chemistry

photocatalysis, co-catalyst, β-FeSe, g-C₃N₄, transfer electrons

**Authors for correspondence:**
Wenwu Zhong
e-mail: tianmenwenwu@163.com
Shangshen Feng
e-mail: fss@tzc.edu.cn

This article has been edited by the Royal Society of Chemistry, including the commissioning, peer review process and editorial aspects up to the point of acceptance.

# β-FeSe nanorods composited g-C₃N₄ with enhanced photocatalytic efficiency

Shijie Shen, Wenwu Zhong, Zongpeng Wang, Zhiping Lin and Shangshen Feng

School of Pharmaceutical and Materials Engineering, Taizhou University, Taizhou 318000, People's Republic of China

SS, 0000-0002-8431-7552; WZ, 0000-0002-9875-7460

A series of β-FeSe nanorods composited g-C₃N₄ were prepared. The structure, morphology, chemical state, photocatalytic activity, electrochemical impedance and photoluminescence of β-FeSe/g-C₃N₄ composites were well characterized. It is found that the decolourization rate of 3 wt% β-FeSe/g-C₃N₄ composites reaches 4.4 times than that of g-C₃N₄. The improved photocatalytic properties could be ascribed to the reduced recombination of photogenerated electrons and holes, which is derived from the excellent ability of β-FeSe to capture and transfer electrons. This work provides an alternative co-catalyst for decolourizing organic matter.

## 1. Introduction

The energy crisis and the increasingly serious environmental pollution problem are the severe challenges facing human survival and development [1–4]. Today, with the depletion of fossil energy, the use of clean solar energy has become an alternative solution. In this regard, photocatalyst plays an important role in harvesting solar energy, which can convert solar energy into chemical energy and use sunlight to degrade organic pollutants. Since the 1970s, TiO₂ has been used as a photocatalyst to split water [5–7]. Nowadays, more semiconductor materials, such as ZnO, [8] SrTiO₃ [9] and CdS, [10] were developed as photocatalysts. Among them, g-C₃N₄ stands out for its wider absorption spectrum and higher efficiency in activating molecular oxygen into superoxide radicals [11–13]. Nevertheless, the performance of g-C₃N₄ is still insufficient for its bulk structure and the high carrier recombination probability. Based on it, there are usually two approaches to enhance the photocatalytic activity, one of which is nanostructure design with more active sites. As far as we know, various nanostructured g-C₃N₄, like nanoparticles, nanospheres, nanorods, nanowires and particularly nanosheets have been developed for photocatalysts with higher

activity [13]. Moreover, g-C$_3$N$_4$ coupled with semiconductors and noble metal nanoparticles can also improve photocatalytic activity [14,15]. In the photocatalytic process, photogenerated electrons and holes undergo two types of reactions, one for driving photocatalytic reactions and the other for recombination. In fact, the latter tends to dominate even though it is harmful to photocatalysis. Therefore, the effective separation of photogenerated carriers to avoid recombination is particularly important. The aforementioned composite materials offer enhanced migration efficiency of photogenerated electrons and holes, and hence suppressed recombination. As for noble metal composites, Au, [16] Ag [17] and Pt [18] have been used to couple with g-C$_3$N$_4$.

FeSe has two crystalline states, a hexagonal phase (α-FeSe) and a tetragonal phase (β-FeSe) [19] Among them, β-FeSe exhibits metallic behaviour above $T_c = 8$ K and becomes a superconductor below that temperature [19]. The room-temperature resistivity of β-FeSe is about 1 mΩ cm [20]. β-FeSe has a layered structure, which consists of a quasi-two-dimensional layer composed of edge-sharing FeSe$_4$ tetrahedra stacking along the $c$-axis. Intercalating metal ions or even neutral molecules into [Fe$_2$Se$_2$] layers will transfer electrons to [Fe$_2$Se$_2$] layer, [21–24] which indicates that β-FeSe has excellent ability to capture electrons. Moreover, β-FeSe nanoparticles, [25] nanoflakes [26] and nanorods [27,28] have been synthesized through various preparation methods.

Considering that the recombination of photogenerated carriers is the main reason for hindering the photocatalytic performance of g-C$_3$N$_4$, the composition of co-catalyst such as semiconductors and noble metal nanoparticles can promote the separation of photogenerated carriers and improve the photocatalytic efficiency. β-FeSe does not contain precious metal elements and is a potential electron capturer as mentioned above. It is a fascinating question whether g-C$_3$N$_4$ composited with β-FeSe nanorods has excellent photocatalytic activity. Here, we prepared a series of β-FeSe nanorods composited g-C$_3$N$_4$ and studied their photocatalytic properties for decolourizing Rhodamine B (RhB). The photocatalytic activity of g-C$_3$N$_4$ is greatly improved after the composition. This work provides a promising co-catalyst for photocatalysis.

# 2. Experimental set-up

## 2.1. Materials

RhB (analytical grade), H$_2$O$_2$ (30 wt%, analytical grade) and Na$_2$SO$_4$ (99%) were obtained from Innochem. Urea (99.999%) and Nafion solution (5 wt%) were provided by Aladdin. Iron pieces (99.99%) and selenium shots (99.999%) were purchased from Alfa Aesar.

## 2.2. Preparation of β-FeSe/g-C$_3$N$_4$ composites

Firstly, g-C$_3$N$_4$ was prepared through high-temperature pyrolysis of urea [29]. About 20 g of urea is contained to the three-quarter height position of the crucible, which was covered by a lid and was sintered at 550°C for 2.5 h. The synthesis of β-FeSe nanorods is described as follows. β-FeSe crystals were synthesized following the method described in [30]. The iron pieces and selenium shots weighed in a nominal ratio were sealed in a quartz tube, which was placed in a muffle furnace at 750°C, kept for 5 days and then heated to 1075°C for 3 days. It was then quickly transferred to a muffle furnace with 420°C, kept for 2 days and then quenched in liquid nitrogen. The obtained β-FeSe crystals were ground into powders using a mortar and then dispersed in absolute ethanol, which was sonicated for 2 h in a high-power ultrasonic instrument. Finally, β-FeSe nanorods were obtained by centrifugal separation. β-FeSe nanorods and g-C$_3$N$_4$ at a ratio of 1, 3, 5 and 10 wt% were dispersed in absolute ethanol. The mixture was ultrasonicated for 2 h to achieve uniform mixing. Then they were dried at 80°C to evaporate the solvent. The obtained sample was further sintered at 150°C for 5 h to get well-joined β-FeSe /g-C$_3$N$_4$ composites.

## 2.3. Characterization

X-ray diffraction (XRD) data were measured by a PANalytical X'pert Pro diffractometer using Cu target radiation. The morphologies were identified by scanning electron microscopy (SEM, Hitachi S-4800). The X-ray photoelectron spectroscopy (XPS) data were collected on a Thermo ESCALAB 250 Xi system. The photoluminescence spectra (PL) were recorded by a Hitachi F-4600 fluorescence spectrometer. The electrochemical impedance spectroscopy (EIS) was measured as follows. Firstly, 5 mg of the 3 wt%

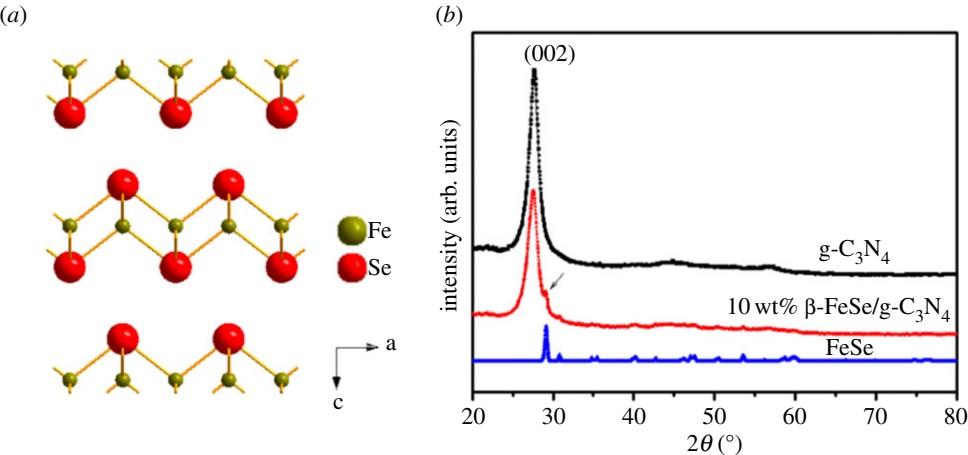

**Figure 1.** (*a*) Schematic diagram of the crystal structure of β-FeSe. (*b*) XRD patterns of as-prepared materials.

β-FeSe/g-$C_3N_4$ composites and 10 µl of 5 wt% Nafion solution were mixed homogeneously in ethanol with 1 ml. The obtained paste was spread on indium tin oxide conductive glass, which was kept at 200°C for 1 h and then used as the working electrode. Moreover, the counter electrode was made of a platinum foil, the reference electrode was made of a saturated Ag/AgCl electrode, and the electrolyte was 0.5 M $Na_2SO_4$ solution. The EIS measurements using the above three-electrode cells were conducted on a CHI 660C electrochemical workstation.

## 2.4. Photocatalytic properties

The photocatalytic properties were characterized by decolourizing RhB on a Shimadzu UV-2450 spectrophotometer using a Xe lamp of 300 W with a filter having a cut-off wavelength of 420 nm under visible light irradiation. In this experiment, 50 mg of β-FeSe/g-$C_3N_4$ composites were added to 50 ml RhB solution, which was stirred continuously for 90 min in dark.

## 3. Results and discussion

The crystal structure of β-FeSe is shown in figure 1*a*. It consists of a quasi-two-dimensional layer composed of edge-sharing $FeSe_4$ tetrahedra stacking along the *c*-axis [19]. The XRD patterns of as-prepared materials are displayed in figure 1*b*. A strong diffraction peak at 27.6°, which corresponds to the (002) reflection, the typical characteristic of g-$C_3N_4$, can be observed [31]. From it, the distance between the stacking layers of the graphitic structure can be derived to be 0.33 nm, which is consistent with the reported g-$C_3N_4$ [32]. The diffraction peaks of β-FeSe can be recognized for 10 wt% β-FeSe/g-$C_3N_4$ composites, as shown in figure 1*b*.

SEM characterization was conducted to obtain the morphology features of the samples. Figure 2*a* displays the SEM image of β-FeSe, from which β-FeSe nanorods could be clearly recognized. The nanorods have diameters of about 30 nm and lengths between 0.3 and 1.2 µm. Figure 2*b* displays the morphology of g-$C_3N_4$. It consists of small pieces of uneven particle size with the order of micrometres (figure 2*b*). As exhibited in figure 2*c,d*, β-FeSe nanorods are dispersed on the outer surface and embedded inside of g-$C_3N_4$.

Figure 3 displays the XPS spectra of 3 wt% β-FeSe/g-$C_3N_4$ composites. The survey spectrum in figure 3*a* displays that there are C, Se, N, O and Fe in the composites. The signals of Fe and Se are weak because of their extremely low content. The peaks of 284.64 and 287.98 eV belong to C 1 s. Among them, the former is ascribable to graphitic carbon [33–35] and the peak located at 287.98 eV is derived from $sp^2$-hybridized carbon (N−C=N) [29]. The spectrum of N 1 s in figure 3*c* can be fitted with three peaks. Among them, the peak of 398.22 eV is due to C=N−C, [36] the peak of 398.80 eV is ascribed to N-$(C)_3$ bond, [36] and the peak located at 400.24 eV corresponds to N-H bond [37]. The Se 3d spectrum (figure 3*d*) consists of two peaks at 55.32 and 59.61 eV for Se $3d_{5/2}$ and $3d_{3/2}$, respectively [27]. The Fe 2p spectrum (figure 3*e*) can be fitted with the peaks at 710.11 and 723.77 eV, which is consistent with the results of Fe 2p in the literature [28,38].

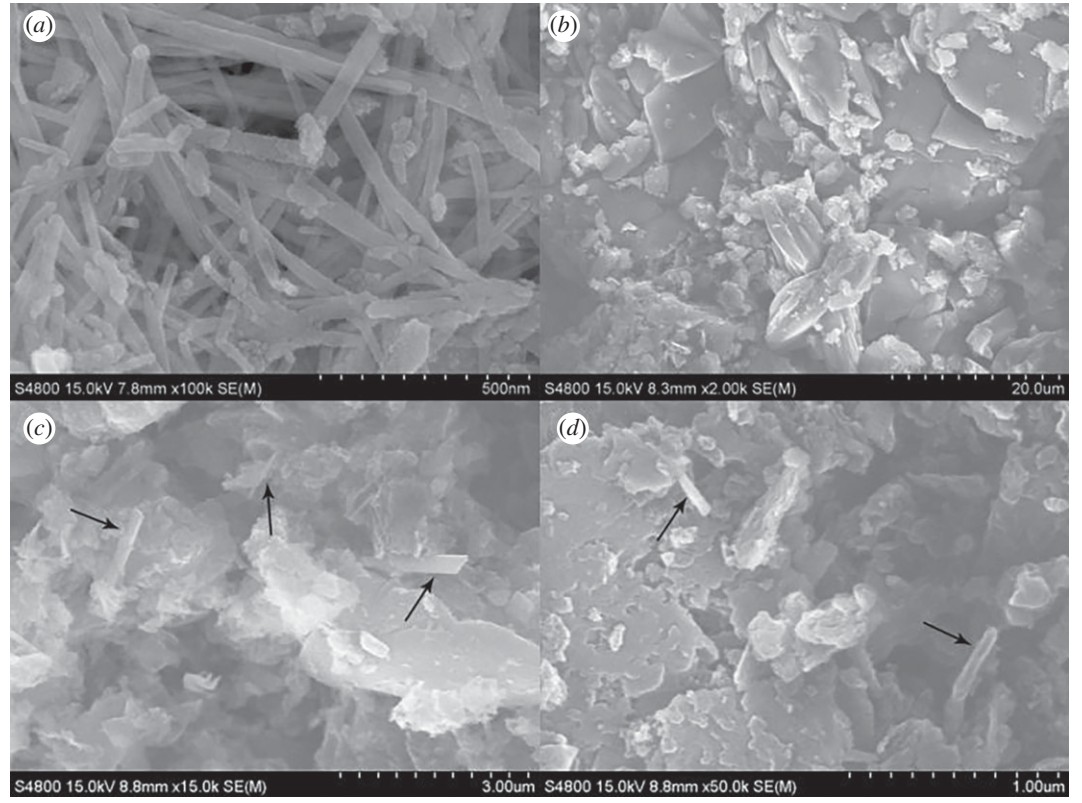

**Figure 2.** SEM images of (*a*) β-FeSe nanorods, (*b*) g-C₃N₄, (*c,d*) 3 wt% β-FeSe/g-C₃N₄ composites.

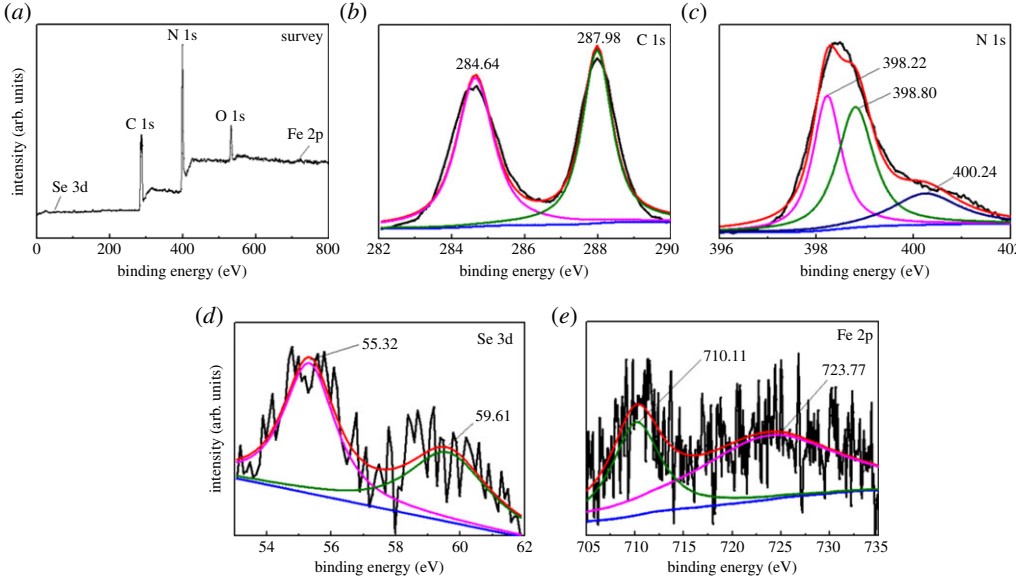

**Figure 3.** XPS spectra of 3 wt% β-FeSe/g-C₃N₄ composites.

The photocatalytic properties of as-prepared materials were evaluated by degrading RhB dyes. Figure 4*a* shows the dark adsorption of RhB on g-C₃N₄ and 3 wt% β-FeSe/g-C₃N₄ composites. It can be seen that RhB is greatly adsorbed by the samples. As shown in figure 4*b*, after exposure to visible light for 180 min, the concentration of RhB was still 90% of the original for g-C₃N₄, while the decolourization efficiency is greatly enhanced for β-FeSe/g-C₃N₄ composites. The optimized ratio is 3 wt% β-FeSe/g-C₃N₄ composites. Further increase in the β-FeSe content leads to a lower decolourization efficiency, which is due to excess β-FeSe shielding the light that reaches the g-C₃N₄ surface and thus affecting the absorption of light. Moreover, the reaction kinetics of RhB

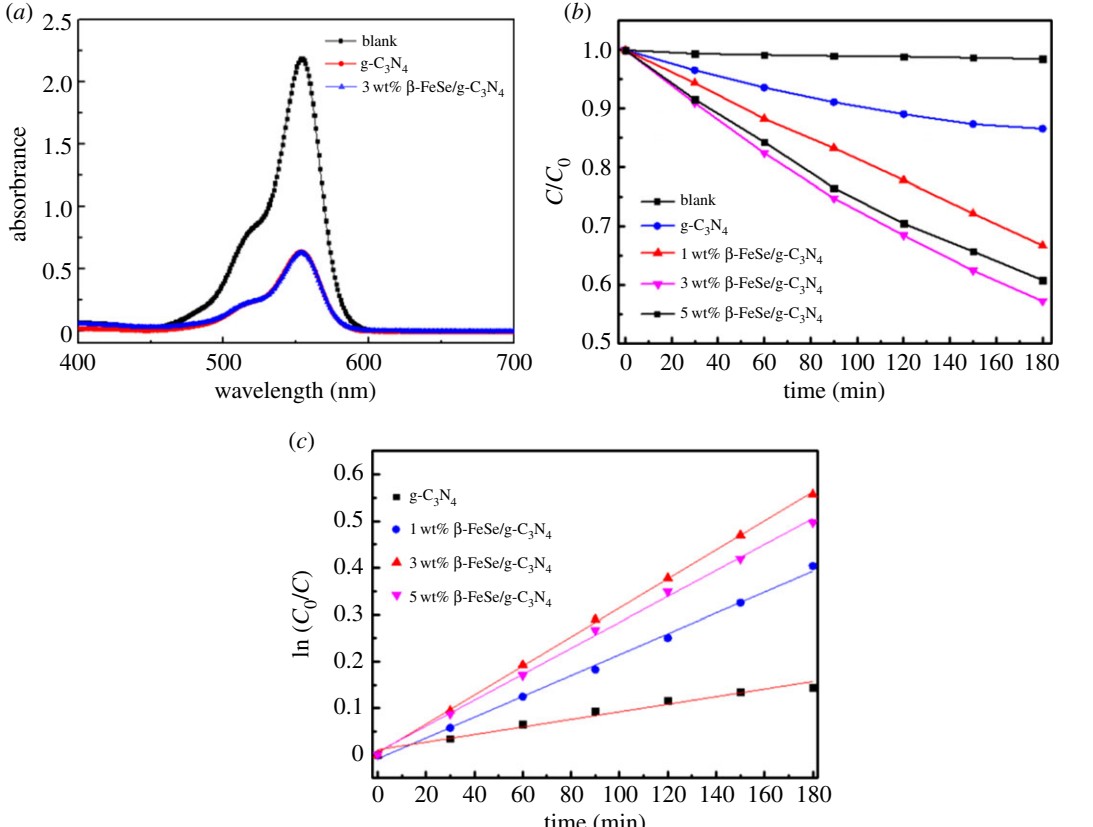

**Figure 4.** (*a*) The dark adsorption of RhB on g-C$_3$N$_4$ and 3 wt% β-FeSe/g-C$_3$N$_4$ composites. (*b,c*) Visible light irradiation photocatalytic activities of as-prepared materials for degrading RhB.

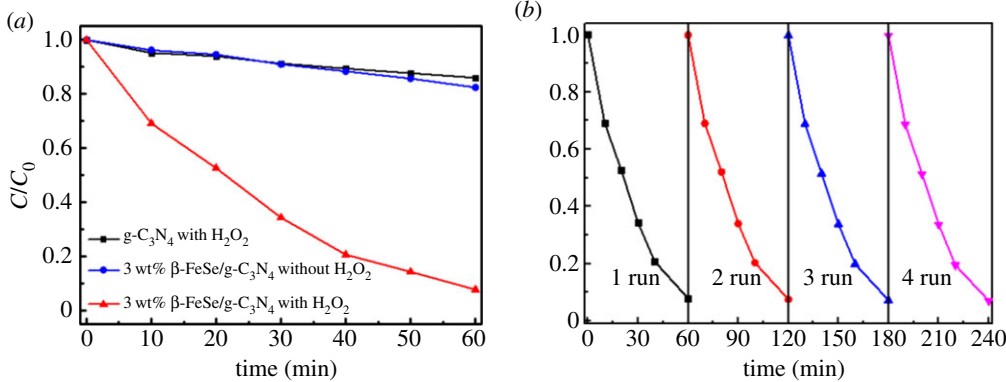

**Figure 5.** (*a*) Photocatalytic activities of decolourization of RhB under visible light for as-prepared materials. (*b*) Cyclic performance of 3 wt% β-FeSe/g-C$_3$N$_4$ composites with H$_2$O$_2$.

decolourization can be fitted by the first-order reaction kinetics ($\ln(C_0/C) = kt$) when $C_0$ is of the order of millimolar, where $C_0$ is the concentration at which RhB reaches the equilibrium of absorption and desorption in the dark, $C$ is the concentration under visible light and $k$ is the first-order reaction rate constant. The results of photocatalytic kinetics are shown in figure 4*c*. The *k*-value is calculated to be 0.00077, 0.0022, 0.0034 and 0.0029 min$^{-1}$ for g-C$_3$N$_4$, 1, 3 and 5 wt% β-FeSe/g-C$_3$N$_4$ composites. So the decolourization rate of 3 wt% β-FeSe/g-C$_3$N$_4$ composites reaches 4.4 times than that of g-C$_3$N$_4$.

To further improve the photocatalytic efficiency, a low concentration of H$_2$O$_2$ (103 μl /100 ml) was used as an efficient scavenger adding in the solution. It can be seen from figure 5*a* that the photocatalytic efficiency of 3 wt% β-FeSe/g-C$_3$N$_4$ composites with H$_2$O$_2$ is greatly improved. The RhB in solution is completely decomposed in 60 min. The improved photocatalytic performance can be

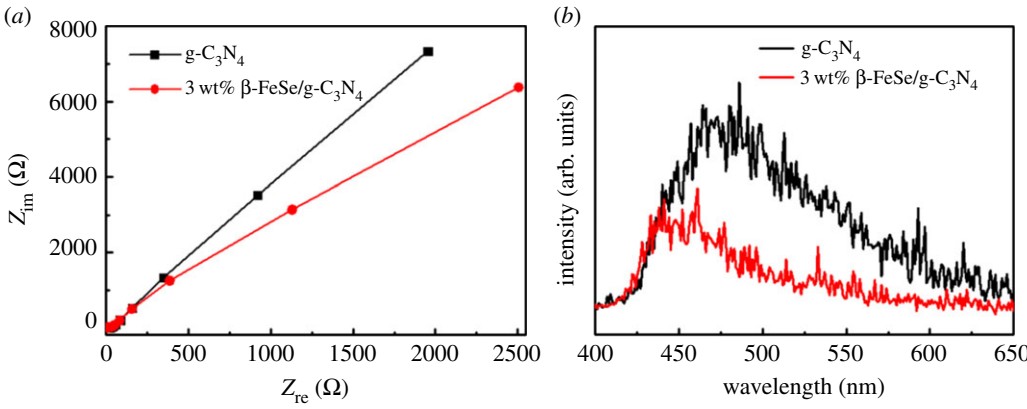

**Figure 6.** (a) EIS at 0.6 V (versus Ag/AgCl) in 0.5 M $Na_2SO_4$ solution under visible light irradiation. (b) PL under 330 nm excitation at 298 K.

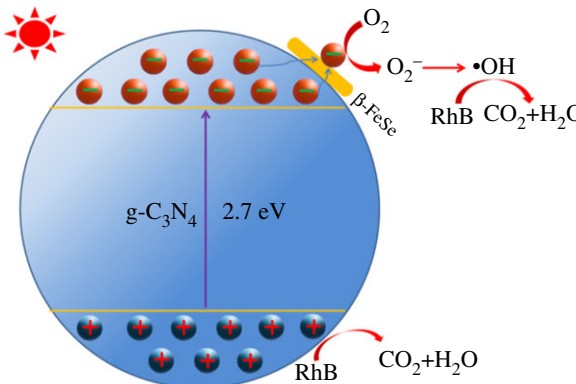

**Figure 7.** Proposed photocatalysis mechanism diagrams of decolourization of RhB for β-FeSe/g-$C_3N_4$ composites.

explained as follows. On the one hand, $H_2O_2$ may undergo photolysis by visible light and generate ·OH radicals, on the other hand, $H_2O_2$ can capture the photogenerated electrons to form ·OH radicals, which are the main contributors to the photocatalytic process [39–41], thus further enhancing photocatalytic activity. The cyclic performance of 3 wt% β-FeSe/g-$C_3N_4$ composites is shown in figure 5b. The photodecolourization rate is almost unchanged after four cycles, manifesting the stability of this photocatalyst.

EIS measurement was performed to obtain the reason for the improvement of photocatalytic performance for 3 wt% β-FeSe/g-$C_3N_4$ composites. As shown in figure 6a, the 3 wt% β-FeSe/g-$C_3N_4$ composites show smaller arc radius than that of g-$C_3N_4$, indicating the former has a smaller charge-transfer resistance than g-$C_3N_4$ and has faster interfacial charge-transfer process [42]. Moreover, photoluminescence spectrum, which is often conducted to understand the charge separation efficiency, [43] was performed. As shown in figure 6b, the intensity of photoluminescence decreases for 3 wt% β-FeSe/g-$C_3N_4$ composites. It manifests that the recombination of photogenerated carriers decreases for the former.

As discussed above, a possible mechanism of improved photocatalytic efficiency for β-FeSe/g-$C_3N_4$ composites is given as shown in figure 7. Under visible light, electrons of g-$C_3N_4$ are excited, which are rapidly transferred to β-FeSe for its excellent ability to capture electrons. The photogenerated electrons then react with $O_2$ to produce superoxide radical anion $O_2^-$. The $O_2^-$ react with water and generate •OH, which degrade RhB to be $CO_2$ and $H_2O$ [39–41]. Moreover, photogenerated holes directly capture the electrons of RhB and discolour it. Since the photogenerated electrons are entrapped and transferred by β-FeSe nanorods, the recombination of the electrons and holes are improved, which was confirmed by the EIS and PL spectra. In other words, the photogenerated electrons and holes involved in the discoloration reaction of RhB are increased. Thus, the photocatalytic efficiency is enhanced for β-FeSe/g-$C_3N_4$ composites.

# 4. Conclusion

In summary, β-FeSe nanorods were used as co-catalyst composited with g-C$_3$N$_4$. The photocatalytic efficiency is remarkably enhanced for β-FeSe/g-C$_3$N$_4$ composites. The decolourization rate of 3 wt% β-FeSe/g-C$_3$N$_4$ composites reaches 4.4 times that of g-C$_3$N$_4$. The RhB in solution is completely decomposed within 60 min for 3 wt% β-FeSe/g-C$_3$N$_4$ composites with H$_2$O$_2$. The photogenerated electrons can be entrapped and transferred by β-FeSe nanorods, which reduces the recombination of the electrons and holes and improves the photocatalytic efficiency. This work provides a promising co-catalyst for photocatalytic discolourization of organic matter.

Data accessibility. Our data are deposited at Dryad Digital Repository: http://dx.doi.org/10.5061/dryad.3310h18 [44].
Authors' contributions. W.Z. and S.F. designed this work; S.S. and W.Z. performed the experiments; Z.W. and Z.L. analysed the data; S.S. wrote this paper. All authors gave final approval for publication.
Competing interests. The authors declare no competing interests.
Funding. Financial support came from National Natural Science Foundation of China (51802211 and 51572183) and Natural Science Foundation of Zhejiang Province, China (LY15E010002).
Acknowledgements. We thank the assistance from Y.C.C. during the experiment.

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
