## [Reviewer comments · Royal Society Open Science]

Review History

RSOS-181886.R0 (Original submission)

Review form: Reviewer 1 (Xiaosong Zhou)

Is the manuscript scientifically sound in its present form?

Yes

Are the interpretations and conclusions justified by the results?

No

Is the language acceptable?

Yes

Is it clear how to access all supporting data?

Yes

Do you have any ethical concerns with this paper?

No

Have you any concerns about statistical analyses in this paper?

No

Recommendation?

Reject

Comments to the Author(s)

It is an interesting study on exploring in β -FeSe nanorods composited g-C₃N₄ with enhanced photocatalytic efficiency. However, some results and discuss are imprecise so that I think a reject is necessary for publication. The main questions and recommendation are listed as follows:

1. The simple combination of β -FeSe and g-C₃N₄ lacks any novelty.
2. The results of XRD for 3wt% β -FeSe/g-C₃N₄ composites, "The diffraction peak of β -FeSe in the composites is almost invisible because of its low content", why not use a higher 5% content? I don't know how the author came to the conclusion that the as-prepared samples is β -FeSe.
3. For XPS spectra of 3wt% β -FeSe/g-C₃N₄ composites, why not provide Fe spectrograms?
4. What role does FeSe play in photocatalytic processes? The author has no reasonable explanation.

Review form: Reviewer 2

Is the manuscript scientifically sound in its present form?

Yes

Are the interpretations and conclusions justified by the results?

Yes

Is the language acceptable?

Yes

Is it clear how to access all supporting data?

Yes

Do you have any ethical concerns with this paper?

No

Have you any concerns about statistical analyses in this paper?

No

Recommendation?

Major revision is needed (please make suggestions in comments)

Comments to the Author(s)

General remarks

I have carefully read the manuscript, entitled " β -FeSe nanorods composited g-C₃N₄ with enhanced photocatalytic efficiency". The article aims to enhance the photocatalytic activity of g-C₃N₄ by coupling with the conducting β -FeSe nanorods. The synthesized composite photocatalyst was sufficiently characterized and its photocatalytic activity was assessed using RhB as a model compound. Although the idea and organization of the work are good, the data

presentation is poor and several parts of the manuscript needs to be rewritten to be clear. This paper might be considered for publication after extensive revision.

Specific observations

The title: Correct the word “efficiency” it should be “efficiency”.

Page 2 line 57: This paragraph should highlight the novelty of the present work. It also, may include the aim of coupling β -FeSe and g-C₃N₄.

Page 2 line 58: Please identify the RHB at the first incidence.

Page 2-3 lines 58- 10: The following sentences are repeated in the abstract and conclusions section. “The photocatalytic rate of 3wt% β -FeSe/g-C₃N₄ composites reaches 4.4 times than that of g-C₃N₄. The improved photocatalytic properties could be ascribed to the reduced recombination of photogenerated electrons and holes, which results from the excellent ability of β -FeSe to capture and transfer electrons. This work provides an alternative co-catalyst instead of noble metals for photocatalysis”.

Page 3 line 14: The experimental section should include a material or chemicals subsection that list all the chemicals used in the study.

Page 3 line 24: Give a brief description of the synthesis of β -FeSe crystals.

Page 4 line 14: Since there is no mineralization test in this study, change the word “degrading” to be decolorization in the whole manuscript.

Page 4 line 23: Delete the sentence “It should be stated that no permissions were required prior to conducting the above research”.

Page 6 line 52: Fig. 4a should include the results of RhB photolysis. Also, authors should include the results of the dark adsorption of RhB on g-C₃N₄, β -FeSe and their composite in the manuscript.

Page 6 line 53: The sentence “As shown in Fig. 4a, after exposure to visible light for 180 minutes, the concentration of RhB was still 90% of the original” should be rewritten to clarify that this is for g-C₃N₄.

Page 7 line 6: Change “ $\ln (Co/C) = kt$ ” to “the first-order reaction kinetics ($\ln (Co/C) = kt$)”.

Page 7 line 42: What is the concentration (purity) of the H₂O₂.

Page 7 line 44: Scavenger for what? Also, what do the authors excluded the probability that H₂O₂ undergoes photolysis by UV light and generate free radicals as follows:

Thus, H₂O₂ can serve as a reservoir of more active reactive oxygen species that would impact the mechanism of photocatalysis.

Page 7 line 46: Fig. 5a should include the results of the 3wt% β -FeSe/g-C₃N₄ without the H₂O₂.

Page 7 line 51: the authors state that “H₂O₂ can capture the photogenerated electrons to form OH radicals, which are the main contributors to the photocatalytic process, thus further enhancing photocatalytic activity”. another possibility for enhancing the photocatalytic activity is the photolysis of H₂O₂ by UV light can generate free radicals as stated above.

Page 9 line 14: Please recheck and rewrite the following sentences in clearer way “Moreover, photogenerated holes cannot directly oxidize OH⁻ to •OH for the valence band position of g-C₃N₄ is more negative than 1.99 V of OH⁻/•OH. They directly capture the electrons of RhB. Since the photogenerated electrons are entrapped by β-FeSe nanorods, the transfer and recombination of the electrons and holes are improved, which was confirmed by the EIS and PL spectra. Thus, the photocatalytic efficiency is enhanced for β-FeSe/g-C₃N₄ composites”.

Page 9 line 30: the conclusion section is repetition of the abstract, it should not be like that. Also, the authors claim that the β-FeSe is a low-cost co-catalyst, there is no proof for this assumption in the manuscript keeping in mind that the synthesis of β-FeSe is complex process.

Decision letter (RSOS-181886.R0)

03-Jan-2019

Dear Dr Shen:

Title: β-FeSe nanorods composited g-C₃N₄ with enhanced photocatalytic efficiency
Manuscript ID: RSOS-181886

The editor assigned to your manuscript has now received comments from reviewers. We would like you to revise your paper in accordance with the referee and Subject Editor suggestions which can be found below (not including confidential reports to the Editor). Please note this decision does not guarantee eventual acceptance.

Please submit your revised paper before 26-Jan-2019. Please note that the revision deadline will expire at 00.00am on this date. If we do not hear from you within this time then it will be assumed that the paper has been withdrawn. In exceptional circumstances, extensions may be possible if agreed with the Editorial Office in advance. We do not allow multiple rounds of revision so we urge you to make every effort to fully address all of the comments at this stage. If deemed necessary by the Editors, your manuscript will be sent back to one or more of the original reviewers for assessment. If the original reviewers are not available we may invite new reviewers.

Once again, thank you for submitting your manuscript to Royal Society Open Science and I look

forward to receiving your revision. If you have any questions at all, please do not hesitate to get in touch.

On behalf of the Subject Editor Professor Anthony Stace and the Associate Editor Professor Claire Carmalt.

RSC Associate Editor:
Comments to the Author:
(There are no comments.)

RSC Subject Editor:
Comments to the Author:
(There are no comments.)

Reviewers' Comments to Author:
Reviewer: 1

Comments to the Author(s)

It is an interesting study on exploring in β -FeSe nanorods composited g-C₃N₄ with enhanced photocatalytic efficiency. However, some results and discuss are imprecise so that I think a reject is necessary for publication. The main questions and recommendation are listed as follows:

1. The simple combination of β -FeSe and g-C₃N₄ lacks any novelty.
2. The results of XRD for 3wt% β -FeSe/g-C₃N₄ composites, "The diffraction peak of β -FeSe in the composites is almost invisible because of its low content", why not use a higher 5% content? I don't know how the author came to the conclusion that the as-prepared samples is β -FeSe.
3. For XPS spectra of 3wt% β -FeSe/g-C₃N₄ composites, why not provide Fe spectrograms?
4. What role does FeSe play in photocatalytic processes? The author has no reasonable explanation.

Reviewer: 2

Comments to the Author(s)
General remarks

I have carefully read the manuscript, entitled " β -FeSe nanorods composited g-C₃N₄ with enhanced photocatalytic efficiency". The article aims to enhance the photocatalytic activity of g-C₃N₄ by coupling with the conducting β -FeSe nanorods. The synthesized composite photocatalyst was sufficiently characterized and its photocatalytic activity was assessed using

RhB as a model compound. Although the idea and organization of the work are good, the data presentation is poor and several parts of the manuscript needs to be rewritten to be clear. This paper might be considered for publication after extensive revision.

Specific observations

The title: Correct the word "efficiency" it should be "efficiency".

Page 2 line 57: This paragraph should highlight the novelty of the present work. It also, may include the aim of coupling β -FeSe and g-C₃N₄.

Page 2 line 58: Please identify the RHB at the first incidence.

Page 2-3 lines 58- 10: The following sentences are repeated in the abstract and conclusions section. "The photocatalytic rate of 3wt% β -FeSe/g-C₃N₄ composites reaches 4.4 times than that of g-C₃N₄. The improved photocatalytic properties could be ascribed to the reduced recombination of photogenerated electrons and holes, which results from the excellent ability of β -FeSe to capture and transfer electrons. This work provides an alternative co-catalyst instead of noble metals for photocatalysis".

Page 3 line 14: The experimental section should include a material or chemicals subsection that list all the chemicals used in the study.

Page 3 line 24: Give a brief description of the synthesis of β -FeSe crystals.

Page 4 line 14: Since there is no mineralization test in this study, change the word "degrading" to be decolorization in the whole manuscript.

Page 4 line 23: Delete the sentence "It should be stated that no permissions were required prior to conducting the above research".

Page 6 line 52: Fig. 4a should include the results of RhB photolysis. Also, authors should include the results of the dark adsorption of RhB on g-C₃N₄, β -FeSe and their composite in the manuscript.

Page 6 line 53: The sentence "As shown in Fig. 4a, after exposure to visible light for 180 minutes, the concentration of RhB was still 90% of the original" should be rewritten to clarify that this is for g-C₃N₄.

Page 7 line 6: Change " $\ln(C_0/C) = kt$ " to "the first-order reaction kinetics ($\ln(C_0/C) = kt$)".

Page 7 line 42: What is the concentration (purity) of the H₂O₂.

Page 7 line 44: Scavenger for what? Also, what do the authors excluded the probability that H₂O₂ undergoes photolysis by UV light and generate free radicals as follows:

Thus, H₂O₂ can serve as a reservoir of more active reactive oxygen species that would impact the mechanism of photocatalysis.

Page 7 line 46: Fig. 5a should include the results of the 3wt% β -FeSe/g-C₃N₄ without the H₂O₂.

Page 7 line 51: the authors state that "H₂O₂ can capture the photogenerated electrons to form OH radicals, which are the main contributors to the photocatalytic process, thus further enhancing

photocatalytic activity". another possibility for enhancing the photocatalytic activity is the photolysis of H₂O₂ by UV light can generate free radicals as stated above.

Page 9 line 14: Please recheck and rewrite the following sentences in clearer way "Moreover, photogenerated holes cannot directly oxidize OH⁻ to •OH for the valence band position of g-C₃N₄ is more negative than 1.99 V of OH⁻/•OH. They directly capture the electrons of RhB. Since the photogenerated electrons are entrapped by β-FeSe nanorods, the transfer and recombination of the electrons and holes are improved, which was confirmed by the EIS and PL spectra. Thus, the photocatalytic efficiency is enhanced for β-FeSe/g-C₃N₄ composites".

Page 9 line 30: the conclusion section is repetition of the abstract, it should not be like that. Also, the authors claim that the β-FeSe is a low-cost co-catalyst, there is no proof for this assumption in the manuscript keeping in mind that the synthesis of β-FeSe is complex process.

Author's Response to Decision Letter for (RSOS-181886.R0)

See Appendix A.

Decision letter (RSOS-181886.R1)

06-Feb-2019

Dear Dr Shen:

Title: β-FeSe nanorods composited g-C₃N₄ with enhanced photocatalytic efficiency
Manuscript ID: RSOS-181886.R1

It is a pleasure to accept your manuscript in its current form for publication in Royal Society Open Science. The chemistry content of Royal Society Open Science is published in collaboration with the Royal Society of Chemistry.

On behalf of the Subject Editor Professor Anthony Stace and the Associate Editor Professor Claire Carmalt.

RSC Associate Editor
Comments to the Author:
(There are no comments.)

Reviewer(s)' Comments to Author:

Appendix A

Dear Editor,

Thanks for your email of 03-Jan-2019 informing us that our work is of potential interest to the readership of your journal. All the comments from the reviewers are highly appreciated. The manuscript is thoroughly revised to further improve the scientific soundness. All revisions are listed below point by point. Hope our revisions are satisfactory to you and the reviewers. Thank you very much for your further consideration.

Yours sincerely,

Wenwu Zhong

Department of Materials

Taizhou University

Taizhou 318000

China

E-mail: tianmenwenwu@163.com

Response to Reviewer 1:

1. The novelty of the present work has been reorganized and added in the introduction part: “Considering that the recombination of photogenerated carriers is the main reason for hindering the photocatalytic performance of g-C₃N₄, the composition of co-catalyst such as semiconductors and noble metal nanoparticles

can promote the separation of photogenerated carriers and improve the photocatalytic efficiency. FeSe does not contain precious metal elements and is a potential electron capturer as mentioned above. It is a fascinating question whether g-C₃N₄ composited with FeSe nanorods has excellent photocatalytic activity. Here, we prepared a series of β -FeSe nanorods composited g-C₃N₄ and studied their photocatalytic properties for decolorizing Rhodamine B (RhB). The photocatalytic activity of g-C₃N₄ is greatly improved after the composition. This work provides a promising co-catalyst for photocatalysis.”

2. A higher 10% content of β -FeSe nanorods is composited with g-C₃N₄ and characterized through XRD. The result is shown in Fig. 1b.
3. The Fe 2p spectrum has been added in the manuscript.
4. An explanation about the role of FeSe plays in photocatalytic processes has been added in the manuscript. “FeSe acts as an electron capturer and transferrer in photocatalytic processes, as shown in Fig.7. Under visible light, electrons of g-C₃N₄ are excited, which are rapidly transferred to β -FeSe for its excellent ability to capture electrons. Since the photogenerated electrons are entrapped and transferred by β -FeSe nanorods, the recombination of the electrons and holes are improved, which was confirmed by the EIS and PL spectra, in other words, the photogenerated electrons and holes involved in the discoloration reaction of RhB are increased. Thus the photocatalytic efficiency is enhanced for β -FeSe/g-C₃N₄ composites.”

Response to Reviewer 2:

1. The word “efficiency” in the title has been corrected to be “efficiency”.
2. (Page 2 line 57) In order to highlight the novelty of the present work and with the aim of coupling β -FeSe and g-C₃N₄, related discussion has been reorganized and added in the introduction part. “Considering that the recombination of photogenerated carriers is the main reason for hindering the photocatalytic performance of g-C₃N₄, the composition of co-catalyst such as semiconductors and noble metal nanoparticles can promote the separation of photogenerated carriers and improve the photocatalytic efficiency. FeSe does not contain precious metal elements and is a potential electron capturer as mentioned above. It is a fascinating question whether g-C₃N₄ composited with FeSe nanorods has excellent photocatalytic activity. Here, we prepared a series of β -FeSe nanorods composited g-C₃N₄ and studied their photocatalytic properties for decolorizing Rhodamine B (RhB). The photocatalytic activity of g-C₃N₄ is greatly improved after the composition. This work provides a promising co-catalyst for photocatalysis.”
3. (Page 2 line 58) RhB at the first incidence has been identified.
4. (Page 2-3 lines 58- 10) The repeated sentences have been revised.
5. (Page 3 line 14) All the chemicals used in the study are added in the experimental section: “RhB (analytical grade), H₂O₂ (30 wt%, analytical grade) and Na₂SO₄ (99%) were obtained from Innochem.

Urea (99.999%) and Nafion solution (5 wt%) was provided by Aladdin. iron pieces (99.99%) and selenium shots (99.999%) were purchased from Alfa Aesar.”

6. (Page 3 line 24) A brief description of the synthesis of β -FeSe crystals is given: “The iron pieces and selenium shots weighed in a nominal ratio were sealed in a quartz tube, which was placed in a muffle furnace at 750 °C, kept for 5 days and then heated to 1075 °C for 3 days. It was then quickly transferred to a muffle furnace with 420 °C, kept for 2 days and then quenched in liquid nitrogen.”
7. (Page 4 line 14) The word “degrading” is changed to be “decolorizing” in the whole manuscript.
8. (Page 4 line 23) The sentence “It should be stated that no permissions were required prior to conducting the above research” is deleted.
9. (Page 6 line52) The results of RhB photolysis have been added in Fig. 4b. The results of the dark adsorption of RhB on g-C₃N₄ and their composite have been added in the manuscript (Fig. 4a).
10. (Page 6 line 53) The sentence “As shown in Fig. 4a, after exposure to visible light for 180 minutes, the concentration of RhB was still 90% of the original” is rewritten to clarify that this is for g-C₃N₄.
11. (Page 7 line 6) “ $\ln (C_0/C) = kt$ ” is changed to “the first-order reaction kinetics ($\ln (C_0/C) = kt$)”.
12. (Page 7 line 42) The concentration of the H₂O₂ is 103 μ L /100 mL. The purity of the H₂O₂ is analytical grade. Related content has been added to manuscript.

13. (Page 7 line 44) H₂O₂ served as a scavenger that could capture the photoinduced electrons to form more ·OH radicals. The probability that H₂O₂ undergoes photolysis by visible light and generate free radicals is added: “On the one hand, H₂O₂ may undergo photolysis by visible light and generate ·OH radicals.”
14. (Page 7 line 46) Fig. 5a is revised to include the results of the 3wt% β-FeSe/g-C₃N₄ without the H₂O₂.
15. (Page 7 line 51) Another possibility for enhancing the photocatalytic activity “the photolysis of H₂O₂ by visible light can generate free radicals” is added.
16. (Page 9 line 14) The related sentences are rechecked and rewritten to be “Moreover, photogenerated holes directly capture the electrons of RhB and discolor it. Since the photogenerated electrons are entrapped and transferred by β-FeSe nanorods, the recombination of the electrons and holes are improved, which was confirmed by the EIS and PL spectra, in other words, the photogenerated electrons and holes involved in the discoloration reaction of RhB are increased. Thus the photocatalytic efficiency is enhanced for β-FeSe/g-C₃N₄ composites.”
17. (Page 9 line 30) The conclusion section has been revised to avoid repetition of the abstract. The relevant wording “low-cost” has been revised.